THE NATURAL HISTORY OF MODEL ORGANISMS

# *Nothobranchius furzeri*, an 'instant' fish from an ephemeral habitat

**Abstract** The turquoise killifish, *Nothobranchius furzeri*, is a promising vertebrate model in ageing research and an emerging model organism in genomics, regenerative medicine, developmental biology and ecotoxicology. Its lifestyle is adapted to the ephemeral nature of shallow pools on the African savannah. Its rapid and short active life commences when rains fill the pool: fish hatch, grow rapidly and mature in as few as two weeks, and then reproduce daily until the pool dries out. Its embryos then become inactive, encased in the dry sediment and protected from the harsh environment until the rains return. This invertebrate-like life cycle (short active phase and long developmental arrest) combined with a vertebrate body plan provide the ideal attributes for a laboratory animal.
DOI: https://doi.org/10.7554/eLife.41548.001

**MARTIN REICHARD\* AND MATEJ POLAČIK**

**\*For correspondence:** reichard@
ivb.cz

**Competing interests:** The
authors declare that no
competing interests exist.

**Reviewing editor:** Stuart RF
King, eLife, United Kingdom

## Introduction

The African savannah is dotted with ephemeral freshwater pools known as water pans, which form during the rainy season. Killifishes of the genus *Nothobranchius*, colloquially called annual fishes, have adapted to live in these pools. Popular with specialist aquarium hobbyists for their stunning colouration, most of 75 described species are available in captivity (*Neumann, 2008*), which is how *Nothobranchius furzeri* first made its way into research laboratories worldwide. Alessandro Cellerino, a physiologist from Scuola Normale Superiore in Pisa, Italy, was intrigued by an incredibly short-lived population of the annual fish that his friend Stefano Valdesalici bred at home. Experimental investigation of its lifespan revealed that the fish matured after just four weeks, after which their mortality increased sharply from the age of six weeks and all the fish died within 10 weeks (*Valdesalici and Cellerino, 2003*). The fish of this strain were named GRZ after Gona Re Zhou National Park in Zimbabwe where the fish were originally collected in 1970 (*Jubb, 1971*). They were later found to live slightly longer as housing conditions developed (*Cellerino et al., 2016*), though it remains the

shortest-lived population of *N. furzeri* yet recorded. Collection trips to Mozambique and Zimbabwe between 2004 and 2016 assembled a set of populations that vary in lifespan and possess a variable level of inbreeding (*Terzibasi et al., 2008*; *Cellerino et al., 2016*; *Reichard et al., 2017a*).

The remarkably short lifespan of *N. furzeri* recapitulates the hallmarks of vertebrate ageing (*Harel et al., 2015*), condensed into few weeks and responsive to pharmacological, dietary and lifestyle interventions (*Cellerino et al., 2016*). Its genome has been sequenced and assembled (*Valenzano et al., 2015*; *Reichwald et al., 2015*), and there are transcriptomes for a number of its tissues (*Petzold et al., 2013*; *Baumgart et al., 2014*; *Baumgart et al., 2017*). Several inbred lines are also available (*Hu and Brunet, 2018*) and genome-editing techniques are widely adopted to rapidly produce stable transgenic lines too (*Harel et al., 2016*). With this background, *N. furzeri* has reached outside the world of ageing research and has become valuable for studies in such disparate branches of investigation as evolutionary genomics (*Reichwald et al., 2015*), regenerative medicine

(*Wendler et al., 2015*), developmental biology (*Hu and Brunet, 2018*) and ecotoxicology (*Philippe et al., 2018*). Here, we explore how our understanding of the natural history of the species makes a key contribution to the development of this species as a laboratory model.

## Natural life cycle and longevity

The life cycle of *N. furzeri* is adapted to the transient nature of its habitat and is strictly separated into two distinct phases. The first phase starts when fish hatch after their pool is inundated with rainwater and immediately start feeding. Explosive juvenile growth brings them to sexual maturity in as little as two weeks, growing from the initial size of 5 mm to 30–50 mm over that period (*Vrtílek et al., 2018a*). Growth slows thereafter as resources are diverted to reproduction, but a body size of 75 mm can be reached in males after 10–15 weeks (*Blažek et al., 2013*; *Vrtílek et al., 2018a*). Importantly, their growth rate is strongly dependent on population density and food availability; fish in dense populations are typically small and sexually immature at an age of 3–5 weeks, both in the wild and in the laboratory (*Graf et al., 2010*; *Grégoir et al., 2018*; *Philippe et al., 2018*; *Vrtílek et al., 2018a*). Individual fish vary in growth rates (*Blažek et al., 2013*), perhaps reflecting variation in personality traits (*Thoré et al., 2018*). Upon sexual maturity, females lay eggs daily. A typical fecundity of 20–120 eggs per day was estimated in wild populations, with more variation among populations than among females within a sample (*Vrtílek and Reichard, 2016*; *Vrtílek et al., 2018b*), seemingly in response to resource availability.

All adult fish die when their pool desiccates, but mortality is strong over the entire lifespan (*Vrtílek et al., 2018c*). In longer-lasting pools (persisting more than four months), fish often disappear despite the habitat still appearing capable of supporting them. The fish likely succumb to a combination of predation by birds, bouts of extremely high water temperatures, decreased water quality from organic input by visiting animals, low dissolved oxygen levels, exhaustion of resources, and physiological deterioration as a trade-off to rapid juvenile development (*Reichard, 2015*). While wild-derived, outbred *N. furzeri* populations have a maximum lifespan (defined as 90% survival at the population level) of 25–42 weeks in captivity

(*Terzibasi et al., 2008*; *Tozzini et al., 2013*; *Blažek et al., 2017*), only one of 13 *N. furzeri* populations monitored in the wild survived beyond the age of 17 weeks and most disappeared at the age of 3–10 weeks as their habitat dried up (*Vrtílek et al., 2018c*). After desiccation, dry sediment retains the eggs in a dormant state (diapause); an adaptation to the harsh environmental conditions, over the entire dry season that lasts 5–11 months (*Cellerino et al., 2016*; *Vrtílek et al., 2018b*).

## Distribution

*N. furzeri* is distributed in southern Mozambique, with a few populations extending to adjacent parts of southern Zimbabwe, only few kilometres across the border (*Figure 1A*). Interestingly, the range of *N. furzeri* lies on a gradient in aridity associated with distance from the coast. Evaporation, the amount of rainfall and its predictability are major environmental factors that vary across that gradient (*Terzibasi et al., 2008*; *Tozzini et al., 2013*; *Polačik et al., 2018*), with the driest and least predictable conditions furthest from the ocean (and at the highest altitude) (*Figure 1A*). *N. furzeri* is absent in the wettest region, where another two *Nothobranchius* species, that otherwise coexist with *N. furzeri*, maintain viable populations. In contrast, *N. furzeri* is the most common *Nothobranchius* species in the dry region, 90–400 km from the coast and 19–220 m above sea level (*Reichard et al., 2017a*). Two isolated Zimbabwean populations are found even higher, at 325–340 m above sea level. To date, we have located and georeferenced 90 *N. furzeri* populations over its entire range and their geographic coordinates are accessible at the Figshare repository (DOI: 10.6084/m9.figshare.7017167). Our sampling was limited by accessibility from roads and there are many more *N. furzeri* populations scattered throughout the Mozambican mopane and miombo woodlands and *Acacia* savannah.

Local pools vary greatly in size and shrink gradually (*Vrtílek et al., 2018b*) (*Figure 2A–B*). Environmental conditions in the pool may be harsh; water temperature can fluctuate between 20 and 36°C from early morning to late afternoon (*Reichard et al., 2009*; *Žák et al., 2018*). Late in the season (in May), morning water temperature can drop to 15°C. Fine sediment is repeatedly disturbed by cattle (which have functionally replaced the wild African megafauna)

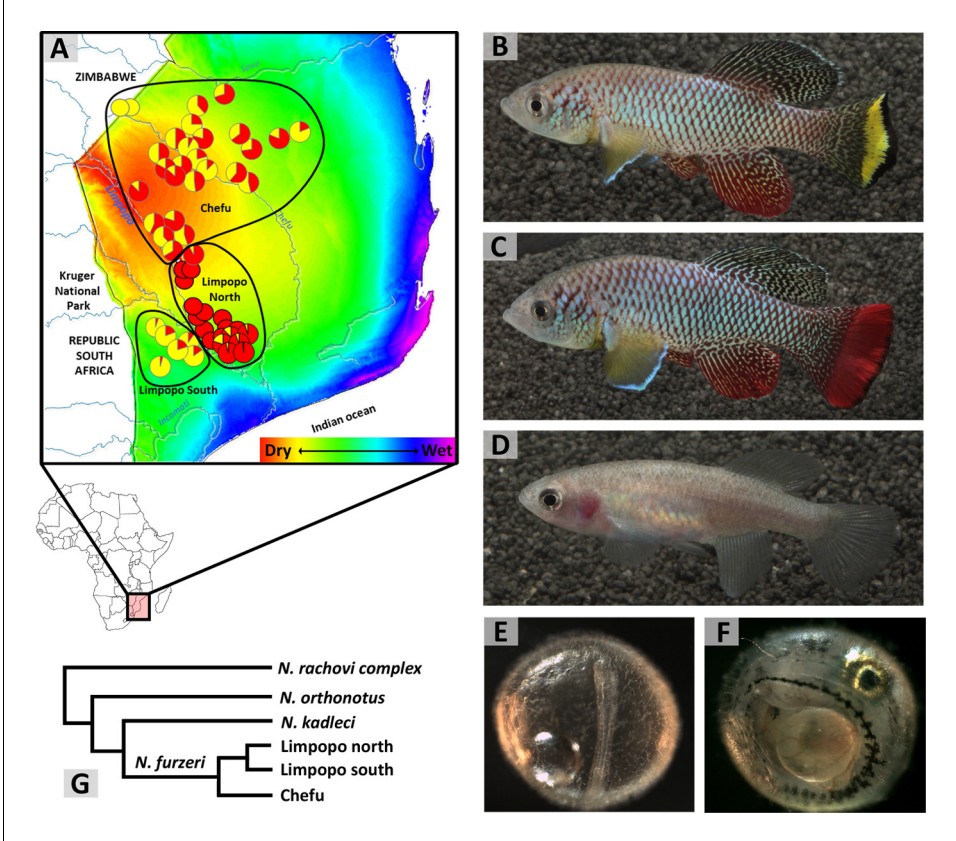

**Figure 1.** Turquoise killifish phenotypes and their distribution. (A) Map of *N. furzeri* distribution across its entire range overlaid on a gradient of aridity (wet to dry: blue to red) and with the proportion of male colour morphs visualised by pie charts and the geographic distribution of intra-specific clades delineated by black lines. (B) Adult yellow morph male, (C) red morph male, and (D) female *N. furzeri*. (E) Embryo at diapause II, representing the longest interval of its lifespan in natural habitats. (F) Embryo at diapause III, fully developed and awaiting hatching cues. (G) Simplified schematic phylogeny of the Southern clade of *Nothobranchius*, with details on *N. furzeri* intra-specific lineages (simplified from *Bartáková et al., 2015*). Image credits: Martin Reichard (1A, 1G), Radim Blažek (1B–1D), Matej Polačik (1E, 1F).

DOI: https://doi.org/10.7554/eLife.41548.002

(*Figure 2C*) and partially dissolves in water, producing extremely turbid conditions (*Figure 2D*). However, in cases when the pool is thickly overgrown with vegetation (*Figure 2E*) and undisturbed by domestic cattle, the water can be transparent.

## Male colour morphs

Two distinct colour morphs of male *N. furzeri* occur, often coexisting in the same population (*Reichard et al., 2009*). The "yellow" morph (*Figure 1B*) possesses a yellow crescent marking along the outer margin of the caudal fin, outlined by a black margin. "Red" morph males (*Figure 1C*) have red caudal fins. The extent of colouration on the caudal fin varies widely among wild populations, but the two morphs are clearly discrete (*Valenzano et al., 2009*; *Ng'oma et al., 2014*). The black margin of the caudal fin is also present in some populations of red males and exceptionally absent in yellow males. Female colouration is dull and brownish and shows no indication of polymorphism (*Figure 1D*).

The geographic distribution of the morphs suggests a link between male colouration and adaptations to different habitat conditions. The yellow morph dominates dry, marginal parts of the range, with exclusively yellow males in the driest region of Zimbabwe. Red males dominate populations in the wet part of the range (*Figure 1A*) (*Reichard et al., 2009*; *Dorn et al., 2011*). This contrast is mirrored in the colouration of the most common laboratory strains, with the short-lived GRZ comprising exclusively

yellow males while the longer-lived MZM0403 are exclusively red (*Cellerino et al., 2016*). Interestingly, most natural populations are polymorphic (*Figure 1A*) and no differences in life history traits between red and yellow males have been established where they coexist. While it is tempting to associate male colouration with adaptation to dry and wet conditions, there is no evidence of any causal link so far.

## Genetic structure

Populations of *N. furzeri* are divided into two main phylogeographic clades (named Chefu and Limpopo), with limited secondary contact (*Terzibasi et al., 2008*; *Bartáková et al., 2013*; *Figure 1A*). The populations are strongly structured, with rare dispersal between them. The population genetic signature reveals that the Chefu clade has recently experienced dramatic population expansion (*Dorn et al., 2011*) and demonstrated linkage between populations along the small rivulet basins (*Bartáková et al., 2015*). Hence, large floods, occurring in exceptionally wet years, are assumed to provide the main mode of dispersal between populations (*Reichard, 2015*). Surprisingly for a fish, large

rivers form boundaries in the distribution of clades and species, including a division of the Limpopo clade to two subclades (*Figure 1G*) and a boundary between the range of *N. furzeri* and its sister species, *N. kadleci* (*Bartáková et al., 2015*).

## Diet

*N. furzeri* are opportunistic and generalist predators of small invertebrates. In general, the diet reflects prey availability in particular pools. Small crustaceans (Cladocera, Copepoda, and Ostracoda) typically constitute more than 70% of digested prey, with insect larvae also being a common prey item (*Polačik and Reichard, 2010*). Backswimmers (nothonectid bugs) are common in the pools but underrepresented in the diet. Chironomid larvae, the most common diet of *N. furzeri* in the laboratory, are also consumed by wild *N. furzeri* (*Polačik and Reichard, 2010*). The diet of *N. furzeri* differs from the two coexisting species (*N. orthonotus* and *N. pienaari*) when prey is abundant though all species resort to any available food of a suitable size when choice is limited (*Polačik et al., 2014a*). The diet is reflected in the intestinal microbiome

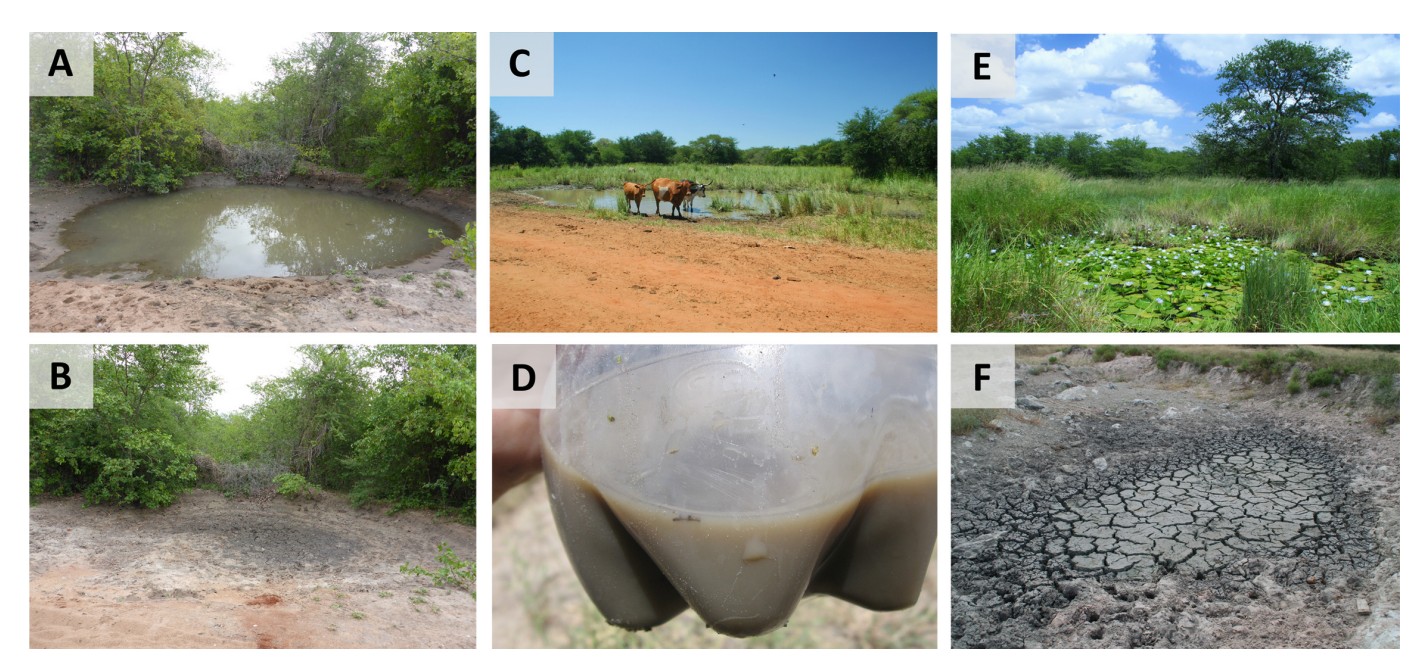

**Figure 2.** Turquoise killifish habitats. (A) A structurally simple habitat one week after filling with water. (B) The same habitat three weeks after filling, fully desiccated. (C) Domestic cattle that commonly visits killifish habitats. (D) Turbid water from a structurally simple killifish habitat discoloured by dissolved fine sediment particles. (E) A structurally complex habitat with abundant aquatic vegetation. (F) Desiccated pool sediment, with typical deep cracks. Image credits: Milan Vrtílek (2A, 2B), Martin Reichard (2C, 2E, 2F), Matej Polačik (2D).
DOI: https://doi.org/10.7554/eLife.41548.003

and wild *N. furzeri* populations differ in the microbial composition of their guts (*Smith et al., 2017*). Interestingly, laboratory fish possess the core microbiota of wild *N. furzeri* and reconstitution of high microbial diversity in old fish extends their lifespan in the laboratory (*Smith et al., 2017*).

## Biotic interactions

### Predators

Potential predators include wading and diving birds, such as herons, hammerheads, storks and kingfishers. These birds prey on killifish, tadpoles and, perhaps, larger invertebrates. Large predatory waterbugs are abundant in many pools and readily consume adult killifish in captivity. These sit-and-wait predators may hunt killifish, especially in pools with dense vegetation. Lungfish (*Protopterus annectens*) commonly coexist with *N. furzeri* and may also prey on them, though they are not their typical predator (*Reichard et al., 2014*). The eggs of *N. furzeri* are likely predated by a range of invertebrates. For instance, freshwater crabs can be abundant in *N. furzeri* habitats and probably extract killifish eggs from the sediment.

### Parasites

The parasites of wild *N. furzeri* have been well characterised (*Nezhybová et al., 2017*). All recorded animal parasites were endoparasites, infecting muscles and internal organs. Perhaps most notably, killifish predominantly hosted intermediate stages of the parasites, forming an important link in the transmission cascade to the final hosts: fish-eating birds. The metacercariae larval stage of parasitic trematodes was by far the most abundant parasite, typically with tens of them infecting the muscle tissue. A surprisingly high diversity of flukes (Trematoda), roundworms (Nematoda) and tapeworms (Cestoda) also reside in *N. furzeri* muscle, intestine, cerebral and abdominal cavities, gallbladder and gills. It is clear that *N. furzeri* are challenged by a multitude of infections over their short lifespan and capable of encysting parasites that penetrated their bodies, providing a potential to study their immune responses.

One fluke species deserves particular attention. Larvae of *Apatemon* sp. infect the cerebral cavity (*Nezhybová et al., 2017*) and manipulate host behaviour. When attacked by a bird, infected killifish stay close to the water surface, even jumping onto water lily pads and thereby exposing themselves to predators. In contrast, uninfected fish quickly seek refuge when exposed to potential predator attack.

## Reproduction and mating behaviour

All *Nothobranchius* species are sexually dimorphic (*Sedláček et al., 2014*). Male colouration and mating behaviour appear strongly sexually selected (*Haas, 1976a*). Males compete aggressively for access to females and large males are dominant (*Polačik and Reichard, 2009*). Females express mate choice by approaching a displaying male. Males actively pursue females and may coerce spawning (*Polačik and Reichard, 2011*). The elaborate male colouration is puzzling, as *N. furzeri* often live in extremely turbid water where visibility is minimal. Indeed, male colouration fades in turbid water. Courtship is brief; the male approaches a female and stops to perform lateral displays with his unpaired fins extended. A receptive female allows the male to fold her in his dorsal fin and the female lays an egg into the sediment. During oviposition, the pair jerks and the sediment is disturbed. With the aid of stiffened rays of the anal fin, the egg is deposited slightly into the substrate (*Passos et al., 2015*). Most eggs are found at pool margins (M. Polačik, unpublished data). There is no consistent choice of red or yellow males by individual *N. furzeri* females (Reichard and Polačik, unpublished data) and mate choice appears to be based on other traits.

Reproduction occurs primarily in the morning. The eggs are ovulated just after sunrise (07:00-08:00) (*Haas, 1976b*) and by 10:00 some females have already spawned all their ovulated eggs. Most females are spent by 15:00 and a new batch of eggs is ovulated next morning (*Vrtílek and Reichard, 2016*). Daily fecundity is strongly contingent upon long-term (the effect of body mass: *Vrtílek and Reichard, 2016*) and short-term (the effect of ration on the preceding day: *Vrtílek and Reichard, 2015*) access to resources.

## Embryo ecology

The embryonic phase of the *N. furzeri* life cycle represents the longest but least understood part of the life cycle in the wild. All *Nothobranchius* populations are limited to pools with a special type of substrate composed of an alkaline swelling clay (*Watters, 2009*), the physical and chemical properties of which are crucial for the survival of *N. furzeri* eggs. Annual killifish

possess three dormancy stages occurring at defined developmental points (*Wourms, 1972*) (*Box 1*). *Watters, 2009* proposed a model of natural embryo development of *Nothobranchius* that links embryo developmental with three phases of incubation conditions (wet, dry and humid) (*Box 1*).

Large variability in embryo developmental dynamics is a key feature for the long-term persistence of *N. furzeri* populations in their highly unpredictable environment (*Furness, 2016*), serving as an effective bet-hedging strategy against false hatching cues, such as insufficient rainfall (*Polačik et al., 2017*). Erratic rains and mid-season desiccation generate conditions for

two (or perhaps even more) generations or cohorts in some years (*Reichard et al., 2017b*).

Laboratory research in *N. furzeri* and other annual killifishes has suggested that developmental disparity is achieved through maternal effects, perhaps based on RNA or protein products. Interestingly, embryos of young females tend to develop rapidly while embryos of older females tend to halt development at diapause (*Podrabsky et al., 2010*; *Polačik et al., 2017*); a seeming adaptation to the possibility of a second inundation in a single rainy season that the eggs of young females experience more often than the eggs of older females (*Polačik et al., 2014b*). However, in the wild the beginning and

## Box 1. Embryo development

Embryonic diapause is a distinctive feature of annual fish. It occurs through a developmental arrest coupled with a marked depression in metabolic rate (*Podrabsky and Hand, 1999*) at three well-defined developmental points called facultative diapauses, all of which can be entered or skipped. Diapause I occurs after an early stage in embryonic development called epiboly, when some cells are distributed across the yolk. It can be induced by a lack of oxygen (anoxia), low temperature and possibly by other conditions (*Wourms, 1972*). Diapause II occurs in the long embryo, midway through embryo development (at the 38 somite stage; *Figure 1E*). Some rudimentary organs are visible but the circulatory system is inactive (*Wourms, 1972*; *Podrabsky et al., 2017*). At diapause III (*Figure 1F*), the embryo has completed its development but awaits appropriate hatching cues at a decreased metabolic rate (*Podrabsky et al., 2010*; *Furness et al., 2015a*; *Furness, 2016*). While this developmental pathway appears strikingly congruent across all annual killifish clades (*Furness et al., 2015b*), it remains to be determined whether it also fully applies to *N. furzeri*. *Watters, 2009* proposed a model of natural embryo development of *Nothobranchius* that links three embryo developmental phases with the three phases of incubation conditions (wet, dry and humid). During the wet phase, the eggs are deposited into the upper substrate layer. Anoxic conditions in the sediment apparently trigger the onset of diapause I (*Watters, 2009*). Any embryo development beyond diapause I in aquatic conditions in the wild is rare (*Domínguez-Castanedo et al., 2013*) and likely linked to embryo position outside the anoxic substrate. The dry incubation phase starts with desiccation (*Figure 2B*), when the substrate loses its moisture and breaks along deep cracks (*Figure 2E*). This oxygenates the substrate and embryos resume their development to reach diapause II (*Watters, 2009*), a stage most resistant to water loss (*Podrabsky et al., 2001*). The egg banks in dry sediment are exclusively composed of diapause II embryos (*Domínguez-Castanedo et al., 2017*). Specific physical features of the clay-rich substrate ensure that some moisture is retained even in the upper substrate layer (*Watters, 2009*). The humid phase starts when the first rains moisten the substrate. Increased moisture and a decrease in oxygen availability supposedly trigger development from diapause II to diapause III (*Watters, 2009*). Embryos at diapause III await a hatching stimulus that is probably associated with torrential rains and inundation of the pool. Hatching is synchronous within and among populations in years when a cyclone-associated rainfall inundates the pools (*Polačik et al., 2011*), but may be asynchronous in dry years (*Reichard et al., 2017b*), perhaps because the substrate is only gradually wetted.

DOI: https://doi.org/10.7554/eLife.41548.004

end of diapause, at least in other killifish species, appears to be controlled by environmental factors (*Matias and Markofsky, 1978*; *Podrabsky et al., 2010*; *Domínguez-Castanedo et al., 2017*) and developmental variability may be achieved through habitat heterogeneity rather than intrinsic control. Gradual desiccation and vertical distribution of embryos create a cline of incubation conditions that generate a staggered embryo development.

The course of embryo development has marked impacts on post-hatching lifespan. Rapidly developing embryos produce phenotypes with a rapid post-hatching strategy, typified by a smaller size at hatching but more rapid growth and sexual maturation, and smaller final size and shorter lifespan (*Polačik et al., 2014b*). It remains to be identified whether embryonic diapause is causally responsible for the limited rapid development later in life, representing one of the outstanding questions to be addressed (*Box 2*). Intriguingly, in a Neotropical annual killifish, insulin-like growth factor signalling regulates developmental trajectories (*Woll and Podrabsky, 2017*), representing candidate regulatory pathway for killifish life history.

## Box 2. Outstanding questions about the natural history of *N. furzeri*

- Do *N. furzeri* senesce in natural populations? In the wild, the life of *N. furzeri* is often brief but may be longer than four months, especially in wet years. How much do wild *N. furzeri* suffer from ageing-related declines?

- How do *N. furzeri* embryos develop under natural conditions? Current knowledge of the embryo development comes from laboratory conditions. As incubation in the laboratory largely differs from natural conditions, we need to know how much it diverges from the natural course of development.

- Which genes underlie lifespan and ageing? With natural populations that predictably vary in lifespan, can we dissect genetic and regulatory pathways associated with differences in ageing?

- Is rapid juvenile development traded off against rapid deterioration later in life? When comparing different *Nothobranchius* species, it appears so, but no longitudinal study has compared life history traits within a single cohort, be it in the laboratory or in the wild.

- Do rapid and slow pace of life coexist within the same *N. furzeri* populations? There is impressive variability in the speed of juvenile development among individuals within the same pool (and laboratory cohort). Is this variation determined by the individual's genetic background or primarily affected by environmental and developmental conditions that an individual experiences during early life?

- Do regulatory pathways that control diapause and ageing overlap? Regulatory networks associated with growth and development are involved in both diapause and ageing in non-vertebrate models (*Frézal and Félix, 2015*). Could that link help us to understand the prevention of ageing-associated damage?

- What are the sources of persistent male-biased mortality in wild populations? Combining data from wild populations with experiments in the laboratory may elucidate why male *N. furzeri*, like many other species including humans, die younger than females.

- Why do males coexist in two colour morphs? Existence of two male morphs is often associated with discrete reproductive tactics or with speciation through adaptation to a divergent environment or from strong female choice.

DOI: https://doi.org/10.7554/eLife.41548.005

## Key advantages of *N. furzeri* for the model species role

The key advantage of *N. furzeri* as a model species stems from the combination of its rapid post-hatching lifestyle and long phase of embryonic arrest that can be altered by manipulating environmental conditions. Having a model with a vertebrate body plan and invertebrate-like life history and lifespan (i.e. short and rapid active lifespan and dormant stage) is a valuable characteristic beyond the field of ageing research, not least because researchers are often constrained by the duration of grant funding. The arrested stage makes the species ideal for storing and shipping and permits flexible time management when working with this 'instant' fish (*Polačik et al., 2016*). In addition, given harsh and variable conditions in natural habitats, captive *N. furzeri* does not require any precise conditions for water quality and housing, perhaps except access to abundant and nutrient rich food (*Polačik et al., 2016*; *Dodzian et al., 2018*).

Information on wild *N. furzeri* populations, and particularly the environmental gradient in its range, have been especially instrumental in explaining differences in lifespan among captive strains (*Tozzini et al., 2013*; *Blažek et al., 2017*) and in validating that its rapid life history is not an artefact of the laboratory (*Vrtílek et al., 2018a*; *Vrtílek et al., 2018c*). Field studies have also strengthened conclusions on the role of microbiota on ageing (*Smith et al., 2017*) and genomic insights into the evolution of short lifespan (*Valenzano et al., 2015*; *Reichwald et al., 2015*). Formulating a standardised artificial diet that would be readily consumed by captive *N. furzeri* now appears critical to increase experimental repeatability across laboratories. Detailed understanding of the natural diet of *N. furzeri* (*Polačik and Reichard, 2010*; *Polačik et al., 2014a*) should help the community to accomplish this goal.

## Conclusions

Compared to other fish model systems in biomedical research (e.g. *Parichy, 2015*), our understanding of the natural history of *N. furzeri* is substantial (*Cellerino et al., 2016*). Indeed, *N. furzeri* may, uniquely, be a model that bridges the interests of ecological and biomedical research (*Cellerino et al., 2016*). Still, we know relatively little about some features of the natural history of *N. furzeri*. While some pressing issues are outlined in *Box 2*, we highlight that the greatest gap in our knowledge of the natural life cycle of *N. furzeri* is in understanding embryonic development under natural conditions. Data from wild populations demonstrated that high phenotypic plasticity of *N. furzeri*, manifested in the laboratory by individually disparate growth rates, timing of sexual maturation and fecundity, is natural. While perhaps complicating some experimental designs, it raises a multitude of questions on the underlying mechanisms, from gene expression to individual behaviour (*Box 2*). With much more to be discovered on the natural history of the species, we anticipate that *N. furzeri* will continue to be instrumental in showing how conceptual and methodological insights from ecological and biomedical research can be integrated within a single research agenda (*Reichard et al., 2015*; *Cellerino et al., 2016*).

## Acknowledgements

We thank R. Blažek, M. Vrtílek, J. Žák, V. Nezhybová and P. Vallo for support during our long-term research on wild killifish populations, R. Blažek, M. Vrtílek, J. Žák, C. Smith, A. Furness, D. R. Valenzano and an anonymous referee for constructive comments on the manuscript, and the Czech Science Foundation for financial support that has kept our killifish research ongoing through several successive projects awarded since 2006.

**Martin Reichard** is at the Czech Academy of Sciences, Institute of Vertebrate Biology, Brno, Czech Republic
reichard@ivb.cz
http://orcid.org/0000-0002-9306-0074

**Matej Polačik** is at the Czech Academy of Sciences, Institute of Vertebrate Biology, Brno, Czech Republic

*Competing interests:* The authors declare that no competing interests exist.

### Funding

| Funder | Grant reference number | Author |
|---|---|---|
| Grantová Agentura České Republiky | 16-00291S | Martin Reichard |
| Grantová Agentura České Republiky | 18-26284S | Matej Polačik |

The funders had no role in study design, data collection and interpretation, or the decision to submit the work for publication.

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
