## [Decision Letter]

Thank you for submitting your article "The Natural History of Model Organisms: *Nothobranchius furzeri*, an 'instant' fish from an ephemeral habitat" for consideration by *eLife*. Your article has been reviewed by three peer reviewers, and the evaluation has been overseen by two Features Editors at *eLife* (Stuart King and Peter Rodgers). The following individuals involved in review of your submission have agreed to reveal their identity: Andrew Furness and Dario Riccardo Valenzano.

After a consultation with the reviewers, the editor has drafted this decision to help you prepare a revised submission.

Summary:

This essay is being considered as part of a series of articles on "The Natural History of Model Organisms" (https://elifesciences.org/collections/8de90445/the-natural-history-of-model-organisms). Each article should explain how our knowledge of the natural history of a model organism has informed recent advances in biology, and how understanding its natural history can influence/advance future studies.

This article summarizes what is known of the natural history of the turquoise killifish (*Nothobranchius furzeri*), an interesting model organism that is gaining traction in several fields of investigation. The manuscript is compelling, clear and easy to read. However, a number of details should be attended to prior to publication.

The editor may also contact you separately about some editorial or stylistic issues that will need to be addressed.

Essential revisions:

1) The sections in the article provide a comprehensive introduction to the natural history of *N. furzeri*. However, more should be done to address "how our understanding of the natural history of the species makes a key contribution to its development as a laboratory model" (as stated in the Introduction). Please revise the sections to emphasise how knowledge of *N. furzeri's* natural history has benefited researchers working with this species in the lab, and/or discoveries in specific fields mentioned in the Introduction. This change would help the article to stand out from other recent (and less-recent) general reviews on *N. furzeri* as a model system in biology (e.g. Terzibasi et al., 2008; Reichard et al., 2015; Cellerino et al., 2016; Hu et al., 2018).

2) Despite the raising interest about *N. furzeri* by the scientific community, the reviewers felt it was an overstatement to refer to it as "an *established* vertebrate model in ageing research" and "*the* species of choice in […] evolutionary genomics, regenerative medicine, developmental biology and ecotoxicology". This text should be toned down. The revised text should aim to give a more balanced view (i.e. pros and cons) of how the natural history of the species contributes to its usefulness as a model organism. (Brief comparisons with other model organisms, including invertebrates and other vertebrates, could be considered to add wider context or strengthen arguments.)

3) Some of the cited work on killifish diapause and reproduction is from species besides *N. furzeri*; yet the way the references are cited does not make this clear to a reader unfamiliar with the system. Please include a 'disclaimer' somewhere in the text to indicate that what is known about diapause/development is drawn from studies of other related annual killifish, and although many aspects seem to be highly congruent among species, it remains to be determined whether the same applies to *N. furzeri* specifically.

4) The main text mentions that little is known about the development in the wild. As such, please consider including at least one or two question about the developmental pathways of *N. furzeri* embryos in the wild into the "Outstanding questions" in Box 2.

5) The paper currently includes 60 references, 28 of which are directly from the two authors. Though the authors have majorly contributed to the development of *N. furzeri* as a biomedical and ecological system, they should consider if all these citations are needed and if other articles have been missed from the reference list.

---

## [Author Response]

Essential revisions:1) The sections in the article provide a comprehensive introduction to the natural history of *N. furzeri*. However, more should be done to address "how our understanding of the natural history of the species makes a key contribution to its development as a laboratory model" (as stated in the Introduction). Please revise the sections to emphasise how knowledge of N. furzeri's natural history has benefited researchers working with this species in the lab, and/or discoveries in specific fields mentioned in the Introduction. This change would help the article to stand out from other recent (and less-recent) general reviews on *N. furzeri* as a model system in biology (e.g. Terzibasi et al., 2008; Reichard et al., 2015; Cellerino et al., 2016; Hu et al., 2018).

We entirely agree with that comment. Ironically, we deleted a special paragraph in the “Key advantages” section in the first version of the manuscript during the final pruning of the text. This section is now included (subsection “Key advantages of *N. furzeri* for the model species role”) and it recapitulates the most instrumental insights that knowledge of natural history of the species benefits lab research. We also included what we see is the current major limitation of *N. furzeri* as a model animal – the lack of artificial standardised diet that would be consumed by adult fish where data from natural populations are crucial. We initially believed that this is outside of the scope of the “natural history” but we now see that lack of this important aspect would be misleading.

In addition, we expanded the section on gut microbiota where that link is very strong (subsection “Diet”) and highlighted the lack of knowledge on immune responses despite the turquoise killifish are faced with multitude of infections in nature and are capable to respond to them immunologically (subsection “Parasites”).

2) Despite the raising interest about *N. furzeri* by the scientific community, the reviewers felt it was an overstatement to refer to it as "an established vertebrate model in ageing research" and "the species of choice in […] evolutionary genomics, regenerative medicine, developmental biology and ecotoxicology". This text should be toned down. The revised text should aim to give a more balanced view (i.e. pros and cons) of how the natural history of the species contributes to its usefulness as a model organism. (Brief comparisons with other model organisms, including invertebrates and other vertebrates, could be considered to add wider context or strengthen arguments.)

We toned down those statements and refer to *N. furzeri* as "promising model in ageing" (Abstract) or "emerging model" (in some other disciplines).

"…become the species of choice…" has been replaced by "…become available for studies…"

We also pay more attention to compare the pros and cons of *N. furzeri* with other model animals, specifically highlighting "invertebrate-like" life history that is beneficial for laboratory breeding and vertebrate body plan (and this rationale is better explained in the current version of the manuscript).

3) Some of the cited work on killifish diapause and reproduction is from species besides *N. furzeri*; yet the way the references are cited does not make this clear to a reader unfamiliar with the system. Please include a 'disclaimer' somewhere in the text to indicate that what is known about diapause/development is drawn from studies of other related annual killifish, and although many aspects seem to be highly congruent among species, it remains to be determined whether the same applies to *N. furzeri* specifically.

Such disclaimer is now included in the main text ("Laboratory research in *N. furzeri* and other annual killifishes has suggested…"; "…diapause, at least in other killifish species, appears to be…") and in Box 1 ("While this developmental pathway appears strikingly congruent across all annual killifish clades (Furness et al., 2015b), it remains to be determined whether it also fully applies to *N. furzeri*.")

4) The main text mentions that little is known about the development in the wild. As such, please consider including at least one or two question about the developmental pathways of *N. furzeri* embryos in the wild into the "Outstanding questions" in Box 2.

That question is now included: "How do *N. furzeri* embryos develop under natural conditions? Current knowledge of the embryo development comes from laboratory conditions. As incubation in the lab largely differs from natural conditions, we need to know how much its diverges from the natural course of development."

5) The paper currently includes 60 references, 28 of which are directly from the two authors. Though the authors have majorly contributed to the development of *N. furzeri* as a biomedical and ecological system, they should consider if all these citations are needed and if other articles have been missed from the reference list.

We are aware of this unfortunate bias. This is primarily driven by the fact that we are indeed the only research group working on the ecology and evolution of wild populations of *N. furzeri* and this review specifically aims at the natural history of the species. We have tried to minimize that bias by including more reference to laboratory studies that investigated aspects of natural history of the species. There are now six new references from other teams added: Grégoir et al., 2017; Thoré et al., 2018, Woll and Podrabsky 2017, Dodzian et al., 2018, and Domiguez-Castanedo et al., 2013; Furness et al., 2015b), although we also added two relevant studies from our team that are highly relevant to the review (on thermoregulation and reproductive senescence in the wild).